# Discrimination of Copper Molten Marks through a Fire Reproduction Experiment Using Microstructure Features

**DOI:** 10.3390/ma15228206

**Published:** 2022-11-18

**Authors:** Jinyoung Park, Joo-Hee Kang, Jiwon Park, Young Ho Ko, Sun Bae Bang

**Affiliations:** 1Korea Electrical Safety Corporation Research Institute, 111, Anjeon-ro, Iseo-myeon, Wanju-gun 55365, Jeollabuk-do, Republic of Korea; 2Korea Institute of Materials Science, 797, Changwon-daero, Seongsan-gu, Changwon-si 51508, Gyeongsangnam-do, Republic of Korea; 3Jeonbuk National University, 567, Baekje-daero, Deokjin-gu, Jeonju-si 54896, Jeollabuk-do, Republic of Korea

**Keywords:** copper wire, molten mark, primary arc bead, secondary arc bead, electron backscatter diffraction, machine learning, fire reproduction experiment

## Abstract

The copper molten marks at a fire site provide important clues for determining the causes of fire. Four factors have been presented to quantitatively discriminate copper molten marks, namely the fraction of (001) component perpendicular to the demarcation line, the grain aspect ratio, the fraction of Σ3 boundaries, and the fraction of maximum grain size. However, only laboratory-level results of these parameters have been presented, and their applicability in actual fires is yet to be verified. In this study, a fire reproduction experimental system was configured to generate molten marks similar to those in actual fire sites. The molten marks were measured by electron backscatter diffraction and applied to the four discriminant factors. The results obtained similar characteristics to those of the laboratory unit, confirming the applicability of the four discriminant factors. Discriminant equations and processes that can distinguish the primary and secondary arc beads were derived using the molten marks generated in the laboratory and reproduction experiments. Furthermore, a probabilistic discrimination method and classification model developed by machine learning were proposed. Therefore, the use of the discriminants in actual fires can improve the reliability of the statistics and prevent the recurrence of similar fires.

## 1. Introduction

Molten marks can be broadly classified into three categories [1,2]. First, a primary arc bead (PAB) refers to a molten bead formed when an electric wire sheath experiences a short circuit due to chemical or thermal deterioration, or an external physical force. PAB also refers to the molten mark that causes a fire. A secondary arc bead (SAB) pertains to the molten mark formed during a short circuit when a copper wire sheath is melted because of a fire. Finally, a globule is a bead formed when an unenergized copper wire is exposed to a temperature higher than the melting point (1083 °C) because of a flame. PABs and SABs are classified as short-circuit marks, which are visually distinguishable from globules [3]; however, distinguishing between PABs and SABs is more complex.

Various equipment and methods have been used to distinguish short-circuit marks that are difficult to identify through eyesight alone. Gray et al. [4] suggested that PABs had square or rectangular pockmarks, which are not as readily identifiable in SABs. Lee et al. [5] distinguished PABs and SABs using the changes in the arm spacing of the dendritic structure according to ambient temperatures when the molten mark was short-circuited. Only PABs contain graphitic carbon in the Raman spectroscopy for the chemical analyses [6]. However, discrimination techniques have not been applied and used in fire investigations. Furthermore, Babrauskas [7] determined that experimental results are not reproducible because it involves subjective judgment and limited scope. Wang et al. [8] proposed a judgement model for identifying the primary and secondary melted marks based on the digital image processed optical micrographs and support vector machine model.

Of the fires in Korea, 9474 (26% of the total fires in 2021) ranked second after careless fires in total fires [9]. In addition, electrical fires are classified into 10 types, with unidentified short circuits accounting for the largest portion (29%). Fires by unidentified short circuits caused about 200 casualties and 58 million dollars of property damage in 2021. An unidentified short circuit refers to a presumed electrical fire caused by a short circuit, whose cause is yet to be identified [10]. These types of fires with unidentifiable causes lower the reliability of fire statistics and cause difficulties in establishing preventive measures, which may result in similar fires. Short-circuit marks are often determined by their appearance only based on the subjective disposition and experiences of the fire investigators at the fire sites. Determination of the cause in unidentified short circuits that exclude the subjective judgement of fire investigators can reduce damage to people and property.

To address this problem, a universal and quantitative method for discriminating molten marks must be proposed. Electron backscatter diffraction (EBSD) analysis is a common method for discriminating molten marks and is used to measure the microtexture and microstructure of almost all metals [11,12]. Compared to the previous works on the molten mark discrimination [Ref Wang] and crack assessment [13] based on the optical micrographs, EBSD analysis has a strong advantage with regard to the direct quantification of crystallographic characteristics. EBSD analysis can be used to interpret the grain boundaries and quantify the crystallographic orientation of each grain. In particular, the microstructure and crystallographic orientation of the molten mark change during solidification because the temperature gradient at the interface between the base wire and melted bead varies under short-circuit and ambient temperature conditions [14]. In other words, PABs were formed by rapid solidification immediately after a short circuit, whereas SABs underwent heat exposure before a short circuit and solidification at a moderate cooling rate, resulting in microstructural changes in their grain shape and orientation. Thus, quantitative EBSD analysis can provide accurate solutions for distinguishing the microstructures of PABs and SABs.

Park et al. [1,2] presented four discriminant parameters that can quantitatively determine molten marks by scanning electron microscopy equipped with EBSD, namely (001)//LD (where LD is perpendicular to the direction of the demarcation line in a molten mark), the grain aspect ratio (GAR), the fraction of Σ3 boundaries (Sig3), and the fraction of maximum grain size to the total molten mark area (GS). Here, (001)//LD indicates the fraction of (001)-oriented grains along the longitudinal direction (LD). The GAR is given by the length of the minor axis divided by the length of the major axis of an ellipse fit to grain. And the GS is the value of maximum grain size (area) divided to the total molten mark area. These four suggested discriminants were derived from laboratory experiments. Therefore, additional validation of the actual fire sites is necessary.

In this study, we confirmed the applicability of these four discriminant factors to actual fires. An actual fire reproduction experimental site was constructed, and molten marks were created by shorting the electrical wires with flames. The discriminant process was derived by combining the molten marks generated in the laboratory and reproduction experiment. The effectiveness of the proposed method was confirmed by collecting the molten marks at the fire site. By applying the four discriminant factors to actual fires, the number of fires caused by unidentified short circuits can be reduced to improve the reliability of the statistics and prevent the recurrence of similar fires.

## 2. Materials and Methods

### 2.1. Experimental Site

A test room was installed at the Gochang Fire Station in Gochang-gun, Jeollabuk-do, South Korea to conduct safe experiments (Appendix A). The internal size of the test room was 3.0 m × 3.0 m × 2.4 m (width × length × height) with a 1.0 m × 2.0 m door at its front. Lumber (0.15 m × 0.15 m × 2.4 m) was placed on the ceiling of the test room, and an insulator was installed at the ends to affix the wire. In addition, a 30 cm × 30 cm open–close door was installed on the upper end of the test room’s side to allow wires to be drawn. Five wooden pallets (1.0 m × 1.0 m × 0.15 m) were stacked in the center of the test room and used as a fire resource.

### 2.2. Electrical Test Configuration

Figure 1 shows the single-wire line diagram of the experimental power supply. The transformer is composed of 300 kVA, 22.9 kV ∆:220/380 V Y. The main circuit breaker is a molded-case circuit breaker (MCCB) with a capacity of 100 A, which was branched into two circuits and connected to a 50 A MCCB, respectively. Ten 30-A earth-leakage circuit breakers were installed in the two terminal distribution boards, and 20 wires were connected to generate a short circuit (CVF 1.5 mm^2^ of two cores).

### 2.3. Instrument

Ambient temperature is a factor that significantly affects the microstructure evolution in copper molten marks; thus, precise temperature measurement is important. K-type thermocouples (temperature range: −230 °C to 1250 °C, limits of error: ±2.2 °C) were installed on the same line as the wire in the east, west, south, north, and center, as shown in Figure 2, to measure the ambient temperature during a short circuit. Appendix A shows the measurement devices. A thermal imaging camera (T530, Flir, Wilsonville, OR, USA) was installed in front of the test room to measure the internal temperature. Four oscilloscopes (WaveRunner 6 Zi, Teledyne LeCroy, Charlottesville, VA, USA) were installed to measure the short-circuit current and voltage. The voltage and current probes were connected to circuits 1 and 10 of distribution panel A, and circuits 1 and 10 of distribution panel B.

### 2.4. Methodology

Ten PABs were fabricated by a short-circuit method at room temperature using an experimental apparatus [1,2] and then exposed by a fire in a test room. The SABs were obtained by installing 20 wires in the test room and shorting them with a flame. Among the 20 copper wires, 15 molten marks were bead-shaped, which were analyzed as SABs. A wood pallet was placed in the center of the test room, ignited with a gas torch and 500 mL gasoline, and exposed to the flame for an additional 5 min after a short circuit. Water was poured to avoid affecting the electric wire when extinguishing the fire. After the short circuit, the molten marks on the electric wires were collected and analyzed.

For revealing of flat and damage-free surface, the molten mark shown was hot-mounted using conductive resin and then wet-ground with SiC papers and polished with diamond suspensions and colloidal silica suspensions. The sample preparation of copper and its alloys for EBSD analysis was introduced elsewhere [1,15,16].

The orientation of the molten marks was obtained using field-emission scanning electron microscopy (FE-SEM; GeminiSEM 560, ZEISS, Obekochen, Germany) equipped with a Velocity Super EBSD camera (EDAX, Pleasanton, CA, USA). An accelerating voltage of 15 kV, and a probe current of 30 nA were used. Mapping was performed on a hexagonal grid with a step size of 4 μm. Post-processing was performed using EDAX OIM 8.6 software (EDAX, Pleasanton, CA, USA) [17] to extract the four discriminant factors. The grain tolerance angle was 5°, and the confidence index (CI) was 0.2 or more. Subsequently, grain CI standardization and neighbor orientation correlation at level 4 was applied.

## 3. Results and Discussion

### 3.1. Temperature Distribution and Molten Mark Appearance

The ambient temperature with the circuit breaker operation was recorded using a camcorder during the short circuit. Figure 3 shows the changes in the temperature over time. A short circuit occurred simultaneously at 240–260 s in all circuits after the flame was applied. At this time, the ambient temperature based on the center thermocouple was 680–750 °C. Appendix A shows capture images during the short circuit (temperature range in the yellow region of Figure 3) at the location with direct flame contact. The maximum short-circuit current flowing through the wire was 400 A.

To observe the appearance of the molten marks, the locations of the short circuit, carbide removal, and the molten marks were obtained to avoid damage. A total of 20 molten marks in the form of beads, notches, and broken strands were observed. Of these, 15 were bead-shaped molten marks. The bead-shaped SABs did not exhibit significant differences in shape from the molten marks generated in the laboratory.

### 3.2. EBSD Microstructure Analysis

Four discriminant factors derived from Park et al. [1,2], namely the fraction of (001)//LD (CD), GAR, fraction of Σ3 boundaries (Sig3), and fraction of maximum grain size to the total molten mark area (GS), could be applied to the molten marks obtained from the fire reproduction experiment. A demarcation line was drawn between the melted and unmelted zones of the copper wire, and the specimen coordinate axis was set as the longitudinal direction (LD), which was perpendicular to the demarcation line [1,2].

Figure 4 and Figure 5 show the LD orientation maps, crystal direction maps, and pole figures (PFs) of three specimens, representing the PABs and SABs. The crystal direction maps (tolerance angle of 15°) show the (001) component (red region), which is (001)//LD. The (001), (011), and (111) PFs were considered for a precise texture analysis. As the PFs express the overall crystal orientation distribution of the molten marks as a density contour, the level of texture development can be checked easily [12]. Trapped air in the molten pool by short-circuit forms pores inside molten marks during rapid solidification.

In cubic metals, <001> is the favorable direction for dendritic growth. Elongated grains parallel to <001> are strongly developed in rapidly cooled microstructures, such as additive-manufactured and fusion welded metals [18,19,20]. Similarly, PABs that underwent rapid solidification after a short circuit had a high fraction of (001)//LD, as shown in Figure 4. Here, LD is the main solidification direction, which has a large thermal gradient during cooling. Although the PABs were exposed to temperatures above 700 °C for approximately 350 s after the short circuit, a high fraction of (001)//LD was maintained. Thus, sufficient recrystallization was obtained at 700 °C. However, the solidified microstructure tends to grow without recrystallization. As a high dislocation density is necessary for recrystallization, the dendritic microstructure formed by solidification has a relatively low strain level. Therefore, the ambient temperature during the short circuit, rather than that after the short circuit, is an important factor in microstructure formation. Based on this result, the microstructure of the short-circuited PABs at room temperature did not change significantly below the melting point of Cu (1083 °C). The (001) fiber texture (Figure 4a) and (rotated) cube texture (Figure 4b,c) are clearly visible in the orientation maps. Moreover, the (001)//LD texture was developed.

As shown in Figure 5, SABs had a low fraction of (001)//LD and weakened texture. In addition, the SAB microstructure became more equiaxed than that of the PABs. SABs were formed when the copper wires exposed to flames undergo recrystallization and grain growth before the short circuit. As the wires were short-circuited at high temperatures, the thermal gradient of SABs was more gradual than that of PABs. In addition, heat dissipation occurred in different directions, thereby diversifying and suppressing the direction of the columnar grains, and distributing the coarse and equiaxed microstructures [21,22]. Therefore, unlike PABs with columnar grains and developed (001)//LD, SABs have a different microstructure.

Figure 6 shows the GAR maps of the PABs and SABs. GAR is given as a value between 0 and 1. As it approaches 0, it indicates an elliptical shape, such as columnar grains. As it approaches 1, it indicates a circular shape, such as equiaxed grains. The PABs short-circuited at room temperature had a large temperature gradient, resulting in the uniform growth of columnar structures in the direction opposite to the heat flow [23,24]. In comparison, the SABs short-circuited at high temperatures have a relatively small thermal gradient, resulting in the dispersed heat flow and slower solidification rate. Moreover, columnar and equiaxed grains coexisted in various directions [23,24].

Figure 7 shows the Σ3 boundary distribution (red line) in the PABs and SABs. The Σ3 boundaries were mainly distributed near the demarcation line, which was more developed in SABs than in PABs. The Σ3 boundary progression is an annealing characteristic. Therefore, the fraction of Σ3 boundaries in PABs was lower because of their high-temperature gradient and suppressed annealing when short-circuited at room temperature [2,25].

Figure 8 shows the comparison of the grain size distribution of PABs and SABs based on the grains with the maximum size. The number of grains with the maximum size in SABs was clearly larger than that of PABs. Under external heat, SABs have a lower thermal gradient, resulting in a lower nucleation rate and grain growth under high temperatures.

Table 1 shows the results of applying the four discriminant factors to the molten marks in the fire reproduction experiment. The average values of GAR, Sig3, and GS factors were higher in SABs, whereas that of the (001)//LD ratio was higher in PABs. This result is consistent with the analysis results of the molten marks generated in the laboratory. Moreover, similar results were obtained for the microstructures of the molten marks generated in the laboratory and fire reproduction experiment. As the fire reproduction experiment was a simulation of an actual fire, we applied and validated the four discriminant factors to an actual fire.

### 3.3. Classification in the Decision Tree

The constructed dataset contained 40 laboratory-generated [2] and 25 molten marks generated by the fire reproduction experiments. Based on the four discriminant factors, a decision tree classification model was built using the scikit-learn Python package [26]. The decision tree model is comprised of simple and interpretable decision rules, and classifies data using Boolean logic. Therefore, it provides discrete classification boundaries for each discriminant factor. The model splits the nodes toward the maximum information gain, which is the difference in the impurity functions between the former and later nodes. The impurity function can be either Gini or Entropy. Gini measures the probability of incorrectly labeled cases, as follows:(1)$$H\left(Q\right)=\sum_{k}p_{k}\left(1-p_{k}\right).$$

Meanwhile, Entropy measures the disorder of the node, as follows:(2)$$H\left(Q\right)=-\sum_{k}p_{k}\log p_{k},$$
where $p_{k}$ is the proportion of class-*k* observations, which are PABs or SABs. In this study, the Gini criterion was selected because it incurs a lower computational cost. In each decision node, it splits toward lowering Gini impurity.

The entire dataset (65 data) was used to train the model, instead of the conventional train/test-set splitting method in machine learning. The model was then tested with four actual fire cases. Figure 9 shows the tree plot, and decision surfaces on the pairs of discriminant factors, whereby only three factors, namely Sig3, GS, and CD, were needed for complete classification. With a criterion of a Sig3 value larger than 3.529, 29 out of 35 SABs were obtained at the root node. Within the three depths, all nodes reached the leaf, indicating complete classification. The results are well projected on the decision surfaces in Figure 9b, particularly on the Sig3–GS pair surface.

The major disadvantage of the decision tree is the sensitivity of the model to data variations. In particular, the tree can be overfitted or becomes unstable, thereby generating different trees depending on the data feed. Nonetheless, this can be mitigated using the ensemble method and hyperparameter tuning with sufficiently large training data. As neither strategy is applicable with limited experimental data, GAR was retained as the discriminant factor for further usage. The decision tree based on 65 datasets was overfitted based on the validation with four actual fire cases, as will be discussed in Section 3.5.

### 3.4. Linear Discriminant Analysis and Process

The linear decision boundary, which can distinguish PABs from SABs by applying a linear discriminant analysis classifier [27], was suggested, as:(3)$$S_{12}={\left[\Sigma^{-1}\left(\mu_{1}^{T}-\mu_{2}^{T}\right)\right]}^{T}x-0.5\Sigma^{-1}\left(\mu_{1}^{T}\mu_{1}-\mu_{2}^{T}\mu_{2}\right)+\ln\frac{P\left(w_{1}\right)}{P\left(w_{2}\right)},$$
where $\mu_{1}$ and $\mu_{2}$ are the average of PABs and SABs, respectively, and Σ^−1^ is the inverse of the common-covariance matrix. To identify the area belonging to PABs and SABs, the text sample (x) can be substituted into *S*_12_ and checked using Equation (4):(4)$$[\;\;\;\;\begin{array}{c} +ve,\;if\;S_{12}>0\;\to x\in PAB\\ sgn\left(S_{12}\right)=0,\;if\;S_{12}=0;on\;the\;boundary.\\ -ve,\;if\;S_{12}<0\;\to x\;\in SAB \end{array}$$

A decision boundary that distinguishes PABs and SABs was obtained by combining two or three discriminant factors. The discriminant accuracy according to each combination is shown in Figure 10. Among them, Sig3–GS–CD (100%) and Sig3–CD–GAR (96.92%) have a higher discriminant accuracy of 95% or higher, as shown in Figure 11a,b, respectively. A discriminant process was subsequently derived (Figure 12).

### 3.5. Application of the Discriminant Method for Molten Marks Identified at the Fire Site

The test samples collected at the fire site were analyzed using EBSD to derive the four discriminant factor values. By substituting the Sig3–GS–CD and Sig3–CD–GAR combinations, which have a discriminant accuracy of 95% or more, the area can be determined, and the posterior probability can be derived. The posterior probability ($p(\omega =\omega_{i}|x)$) can be expressed as a proportional relationship, as shown in Equation (5), using the priori probability ($p(\omega_{i}$)) and conditional probability density function ($p(x|\omega =\omega_{i}$)). The posterior probability represents the probability that the test sample (x) belongs to a group of PABs or SABs, as follows [27]:(5)$$p\left(\omega =\omega_{i}|x\right)\propto p\left(x|\omega =\omega_{i}\right)p\left(\omega_{i}\right).$$
$p\left(x|\omega =\omega_{i}\right)$ is normally distributed ($p\left(x|\omega_{i}\right)\;\sim \;N\left(\mu_{i},\;\Sigma_{i}\right)$, and can be expressed as:(6)$$p\left(x|\omega_{i}\right)=N\left(\mu_{i,\;}\Sigma_{i}\right)=\frac{1}{\sqrt{{\left(2\pi \right)}^{m}\left|\Sigma \right|}}\exp[\frac{-{\left(x-\mu_{i}\right)}^{T}\Sigma^{-1}\left(x-\mu_{i}\right)}{2}],$$
where $\mu_{i}$ is the mean of the PABs or SABs, m is the number of variables in sample (x), and Σ^−1^ is the common covariance matrix.

In the first case, a fire occurred in a textile workshop. A pattern was identified at an outlet installed on a carbonized wall, where the flames spread (Appendix A). It was determined that the fire started from an outlet, and sample A was identified, which was believed to have caused a short circuit based on the expanding flames in the wiring of a nearby light. In the second case, a short circuit occurred in the electrical cord of a washing machine in the multipurpose room of a house. As shown in Appendix A, the short circuit occurred at the position where the electric cord of the washing machine was bent, forming sample B. For the third case, a fire occurred in the dressing room of a manufacturing plant. Appendix A shows that the ceiling experienced a large degree of loss, with a pattern of expanded combustion from the ceiling to the floor. The fire was estimated to start from the ceiling as a molten mark (sample C), which was presumed to be the cause. The fourth case was a self-manufactured molten mark, where the expanding flame caused the short-circuit of the wire after a fire occurred at the outlet, which corresponds to SABs (Appendix A). The ambient temperature at which the short circuit occurred was 634 °C.

Table 2 shows the values of the four discriminant factors, classification results using the decision tree criteria, and posterior probability of samples A–D. Figure 13 presents the position of the samples on the Sig3–GS–CD and Sig3–CD–GAR graphs. The probability of SABs in sample A and D was 99.86% and 89.72%, respectively. In comparison, sample B and C had a PAB probability of 93.84% and 95.37%, respectively. The results of the actual measurements at the fire site matched well, thereby confirming the usefulness of the discriminant factors. The predicted class (PAB) of sample D based on the decision-tree model was different from the actual class (SAB), indicating the overfitting of the model.

## 4. Conclusions

In this study, molten marks were generated through a fire reproduction experiment and applied to obtain the four discriminant factors derived from previous studies, namely (001)//LD (CD), GAR, fraction of Σ3 boundaries (Sig3), and fraction of maximum grain size to the total molten mark area (GS), in order to confirm their applicability to actual fires. A discriminant process was derived using the Sig3–GS–CD and Sig–CD–GAR combinations with a discriminant accuracy of 95% or more. The discriminant factors of the molten marks collected at the actual fire site were applied into the identification process to ensure reliability. The decision tree model successfully classified the PABs and SABs in the experimental molten marks. The tree confirmed the clear judgment criteria for each discriminant factor.

Subsequent studies are expected to expand the discrimination range of the molten marks by analyzing the marks formed using solid copper wires with various diameters and stranded wires and under various conditions, such as overcurrent and tracking.

## Figures and Tables

**Figure 1 materials-15-08206-f001:**
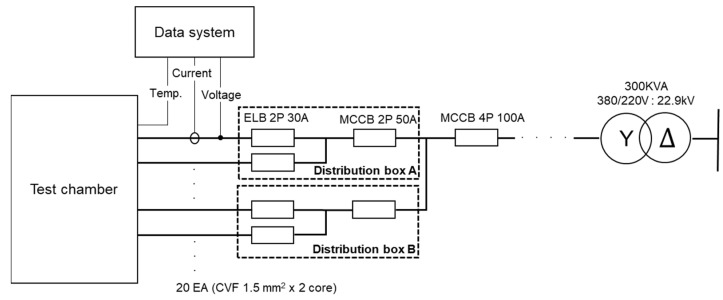
Single-wire line diagram of the experimental power supply. ELB: earth-leakage circuit breaker; MCCB: molded-case circuit breaker.

**Figure 2 materials-15-08206-f002:**
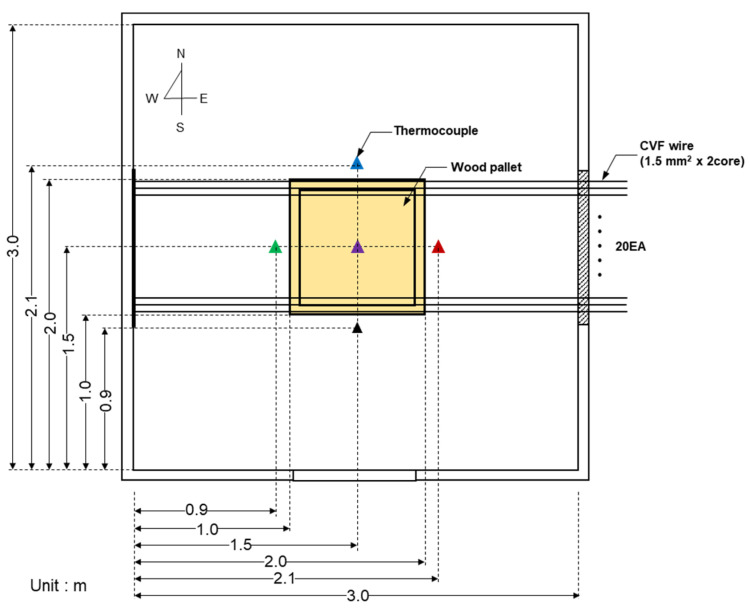
Internal configuration of the test room (top view): thermocouple installed in the east, west, south, north, and center, and wood pallet installed in the center.

**Figure 3 materials-15-08206-f003:**
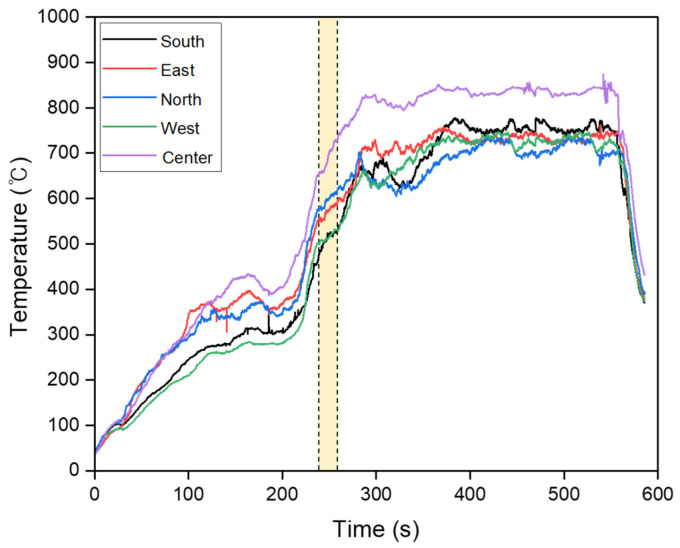
Temperature of the thermocouples over time. All wires were short-circuited within the yellow region (680–750 °C).

**Figure 4 materials-15-08206-f004:**
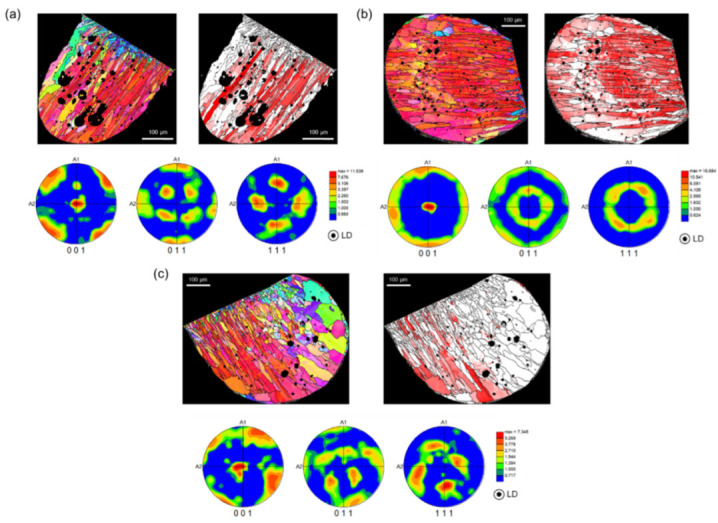
(**a**) Longitudinal direction (LD) orientation maps, (**b**) crystal direction maps of (001)//LD, and (**c**) pole figures of PABs of three representative specimens.

**Figure 5 materials-15-08206-f005:**
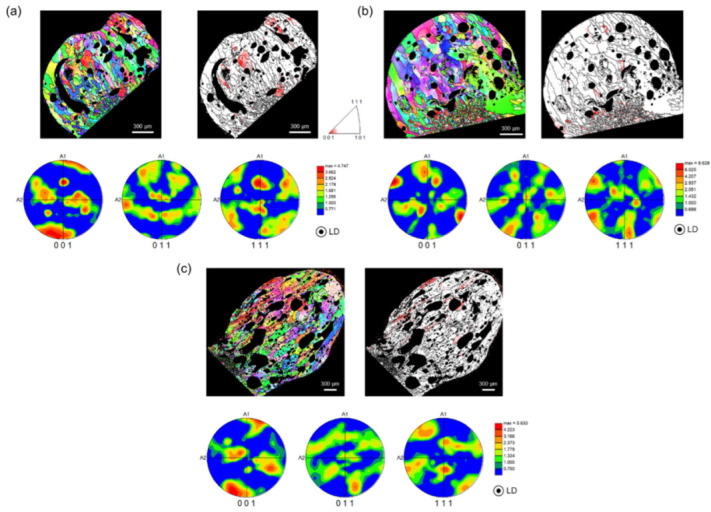
(**a**) Longitudinal direction (LD) orientation maps, (**b**) crystal direction maps of (001)//LD, and (**c**) pole figures of the SABs of three representative specimens.

**Figure 6 materials-15-08206-f006:**
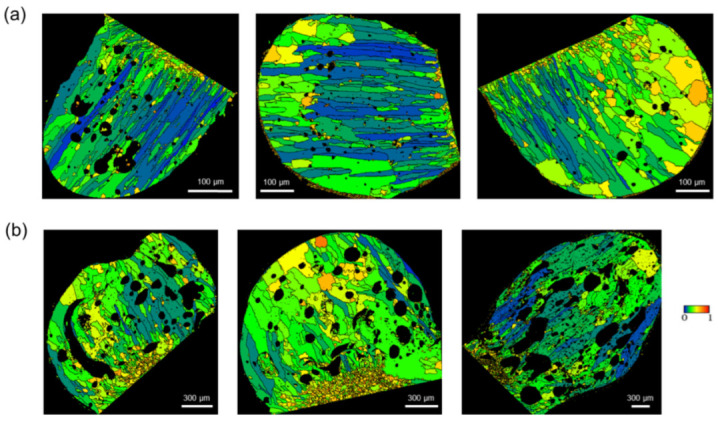
Grain aspect ratio maps: (**a**) PABs and (**b**) SABs.

**Figure 7 materials-15-08206-f007:**
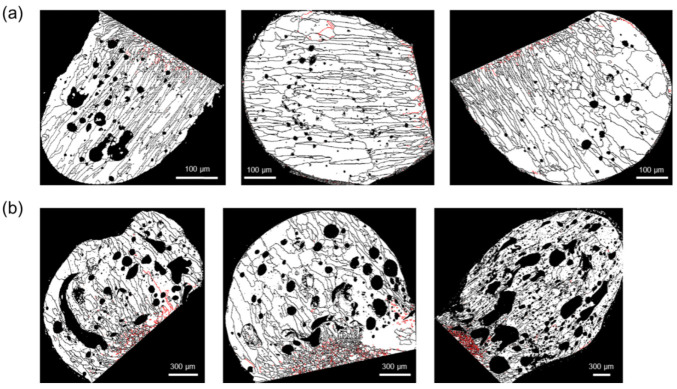
Distribution of Σ3 boundaries (red line) in the molten marks: (**a**) PABs and (**b**) SABs.

**Figure 8 materials-15-08206-f008:**
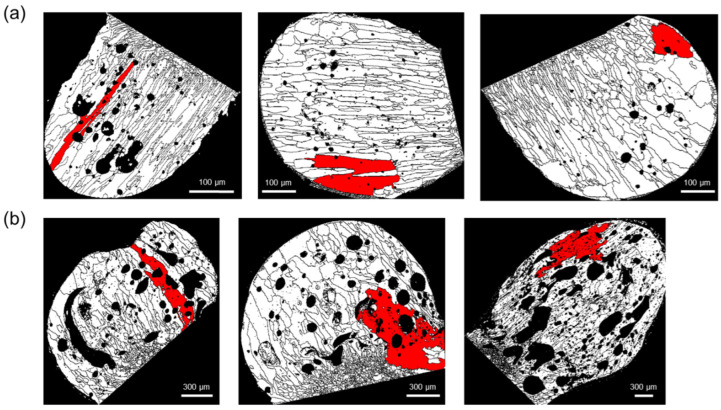
Representation of the grains of maximum size: (**a**) PABs and (**b**) SABs.

**Figure 9 materials-15-08206-f009:**
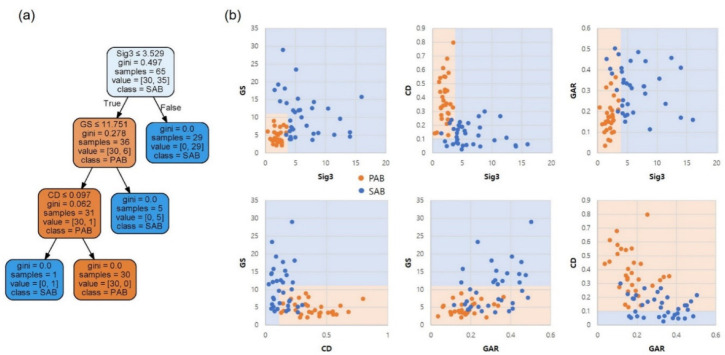
(**a**) Decision tree plot and (**b**) decision surfaces on the pairs of discriminant factors.

**Figure 10 materials-15-08206-f010:**
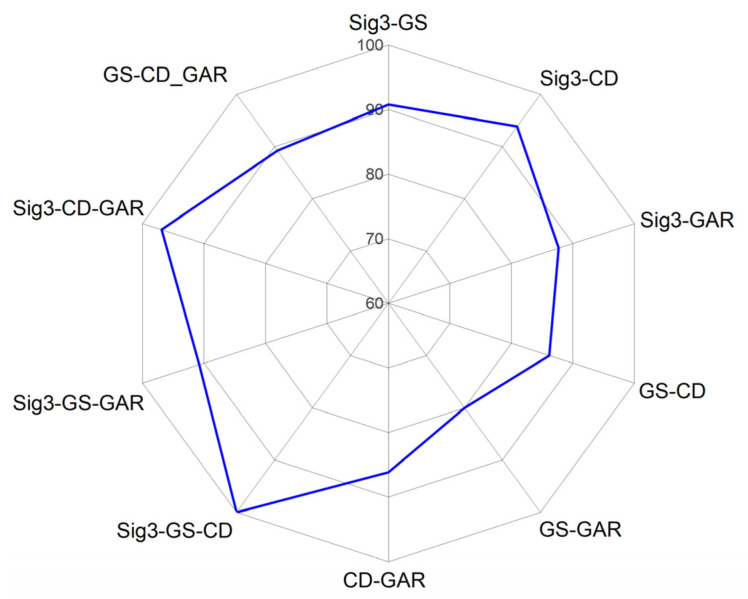
Discriminant accuracy according to the combination of the four discriminant factors.

**Figure 11 materials-15-08206-f011:**
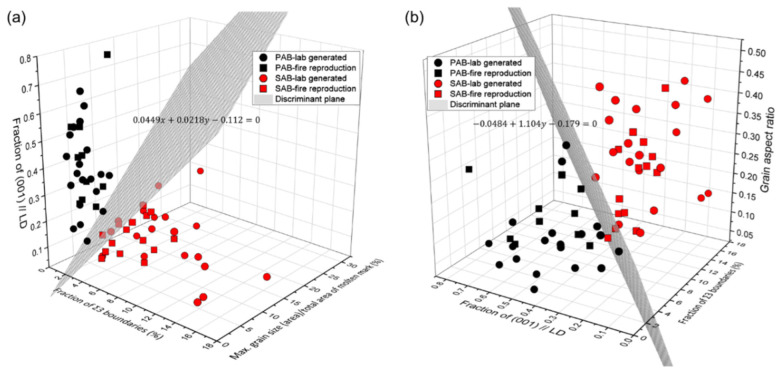
Distribution and decision boundary of PABs and SABs: (**a**) Sig3–GS–CD and (**b**) Sig3–CD–GAR.

**Figure 12 materials-15-08206-f012:**
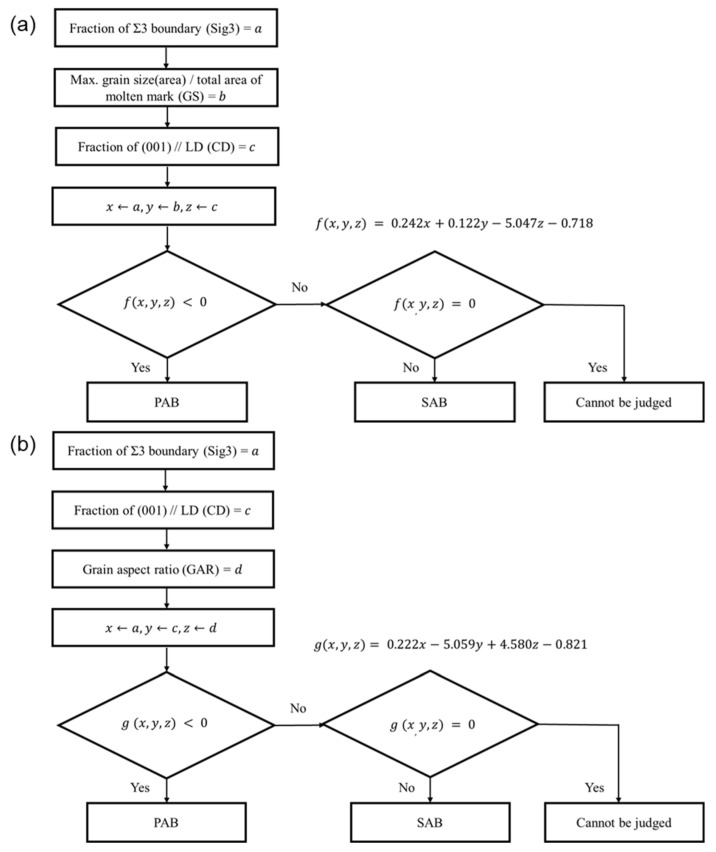
Discriminant process of PABs and SABs: (**a**) Sig3–GS–CD and (**b**) Sig3–CD–GAR.

**Figure 13 materials-15-08206-f013:**
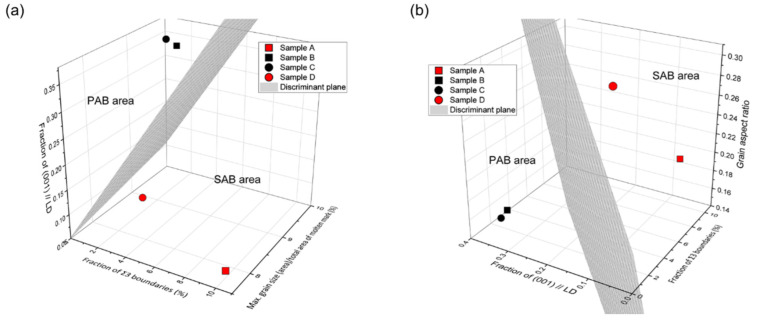
Distribution of the molten marks collected from the fire sites: (**a**) Sig3–GS–CD and (**b**) Sig3–CD–GAR.

**Table 1 materials-15-08206-t001:** Average values of the four discriminant factors.

Molten Mark	(001)//LD	GAR	Sig3 (%)	GS (%)
PAB	0.426	0.205	2.435	4.430
SAB	0.154	0.279	5.677	7.390

GAR, grain aspect ratio; LD, longitudinal direction; PAB, primary arc bead; SAB, secondary arc bead; GS, fraction of maximum grain size to the total molten mark.

**Table 2 materials-15-08206-t002:** Value of the discriminant factors, decision tree predicted classes, and posterior probability of the molten marks at the fire site.

Sample	Sig3	GS	CD	GAR	DT Class	Posterior Probability					
						Sig3–GS–CD		Sig3–CD–GAR		Average	
						PAB	SAB	PAB	SAB	PAB	SAB
A	10.16	7.68	0.072	0.19	SAB	0.06	99.94	0.33	99.77	0.20	**99.86**
B	1.41	9.80	0.33	0.16	PAB	90.16	9.84	97.51	2.49	**93.84**	6.17
C	2.08	9.27	0.37	0.14	PAB	92.49	7.51	98.25	1.75	**95.37**	4.63
D	2.81	8.33	0.11	0.30	PAB	12.19	87.81	8.37	91.63	10.28	**89.72**

## Data Availability

The data related to this work can be obtained from the corresponding authors upon reasonable request.

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
