# Peer review of "Discrimination of Copper Molten Marks through a Fire Reproduction Experiment Using Microstructure Features"

_materials, 2022, doi:10.3390/ma15228206_

Round 1
Reviewer 1 Report
Dear Authors,
Please find comments on the paper “Discrimination of copper molten marks through a fire reproduction experiment using microstructure features”. The paper describes the use of EBSD microstructure analysis to study four factors for a quantitatively discrimination of copper molten marks. The research performed in this work fits the scope of Materials, and it would be of interest to the readers. The manuscript is well written, with essential and high-quality results in the field. Therefore, the manuscript deserves its publication as it is.
Author Response
I appreciated your valuable comments. I attach the responses to reviewer 1.

Reviewer 2 Report
The authors have identified the potential area for conducting this research, and may be useful for practical applications; the following comments shall be addressed
1. Strong background in terms of need, and industry relevance of the study to be discussed
2. The reason for selecting discriminant factors for the result analysis
3. Add the relevant pictures of experiments
4. Limitations and scope for future studies to be discussed
5. Discuss the study parameters such as grain size, aspect ratio little better
6. Add the details of EBSD, sampling and analysis better for reader understanding
7. The relevance and accuracy of using K type thermocouple
8. Some of the technical terms or terminologies to be defined before the explanations in the result section
Refer the following manuscripts for discussion
Wang, L., Liang, D., & Mo, S. (2016). Judgment model for identifying the type of electric molten mark in fire. Journal of Computational Methods in Sciences and Engineering, 16(1), 125-133.
Brooks, C. R., & Choudhury, A. (2002). Failure analysis of engineering materials. McGraw-Hill Education.
Andrushia, A. D., Anand, N., & Arulraj, G. P. (2020). A novel approach for thermal crack detection and quantification in structural concrete using ripplet transform. Struct Control Health Monit, 27(11), e2621.
Author Response
I appreciated your valuable comments. I attach the responses to reviewer 2.

Reviewer 3 Report
This is quite a nice paper. I enjoyed reading it. Major revisions are in order for the authors to address the comments detailed below:
Language needs to be polished. Several mistakes found.
“hat are difficult to identify through eyesight alone.”: why? Are the marks small? Clarify.
“and is used to measure the microtexture and microstructure of almost all metals”: references needed. See for example 10.1016/j.scriptamat.2022.115053 and 10.1016/j.matdes.2022.111176 and revise.
Does the surface preparation required for EBSD influence the observation of the marks? Unclear. Detail please.
Do the authors have an actual photo of the experimental apparatus?
“the fraction of (001)//LD (CD), GAR, fraction of Σ3 boundaries (Sig3), and fraction of maximum grain size to the total molten mark area (GS)”: must be detailed what each of this parameters can provide in terms of information of the molten marks?
Are the pores in the EBSD maps from the fire? Or does the material already contained pores? Clarify this aspect.
“grains parallel to <001> are strongly developed in rapidly cooled microstructures, such as additive-manufactured metals”: and also on fusion welding processes. See 10.1016/j.matdes.2022.110717 and complement.
“Thus, sufficient recrystallization was obtained at 700 °C.”: how much is sufficient? Can this be quantified?
“impurity function can be either Gini or Entropy”: what are the differences?
“ollected at the actual fire site were applied into the identification process to ensure reliability”: was the realibitlity of the proposed method quantified?
Author Response
I appreciated your valuable comments. I attach the responses to reviewer 3.

Round 2
Reviewer 3 Report
Accept